# Positive Interactions between *Aceria pallida* and *Bactericera gobica* on Goji Berry Plants

**DOI:** 10.3390/insects13070577

**Published:** 2022-06-24

**Authors:** Pengxiang Wu, Yang Ge, Jia He, Muhammad Haseeb, Runzhi Zhang

**Affiliations:** 1Key Laboratory of Zoological Systematics and Evolution, Institute of Zoology, Chinese Academy of Sciences, Beijing 100101, China; wupengxiang@ioz.ac.cn; 2College of Life Science, University of Chinese Academy of Sciences, Beijing 100049, China; 3State Key Laboratory Breeding Base of Dao-di Herbs, National Resource Center for Chinese Materia Medica, China Academy of Chinese Medical Sciences, Beijing 100700, China; 13120154491@163.com; 4Institute of Plant Protection, Academy of Ningxia Agriculture and Forestry Science, Yinchuan 750002, China; hejiayc@126.com; 5Center for Biological Control, College of Agriculture and Food Sciences, Florida A&M University, Tallahassee, FL 32307, USA

**Keywords:** *Lycium barbarum* L., Eriophyid mite, psyllid, coexistence, mutualism

## Abstract

**Simple Summary:**

The gall mite *Aceria pallida* can be phoretic on the psyllid *Bactericera gobica* to overwinter. Phoresy is often considered as a pattern of phoront–vector mutualism. The phoront *A. pallida* benefits from phoresy during the overwintering season, but no advantages to the vector *B. gobica* were found during this period. Therefore, this mutualism may occur during the growing season. Because both species share the same host plant and habitat after detachment, interspecific interactions are very likely to occur. To determine whether such interactions were positive or negative, we studied relationships between *A. pallida* and *B. gobica* egg abundances on leaves. Our study suggests positive interactions between two pests during the growing season. Outcomes of positive relationships between gall diameter and mite abundance in the gall provided a way to rapidly estimate mite abundances in the field by measuring gall diameters.

**Abstract:**

The gall mite *Aceria pallida* and the psyllid *Bactericera gobica* are serious Goji berry pests. The mite can be phoretic on the psyllid to overwinter, but it is unclear whether the vector can obtain benefits from the phoront during the growing season. After detachment, the mite shares the same habitat with its vector, so there are very likely to be interspecific interactions. To better understand whether the interactions are positive or negative, information on relationships between abundances of *A. pallida* and *B. gobica* on leaves is needed. Here, *B. gobica* abundance was represented by the egg abundance because the inactive nymphs develop on the same sites after hatching. (1) We found a positive linear relationship between the gall diameter and the mite abundance in the gall (one more millimeter on gall diameter for every 30 mites increase), which provided a way to rapidly estimate mite abundances in the field by measuring gall diameters. (2) There was a positive relationship between the abundance of mites and psyllid eggs on leaves. (3) Both species had positive effects on each other’s habitat selections. More importantly, the interactions of the two species prevented leaf abscission induced by *B. gobica* (leaf lifespan increased by 62.9%), increasing the continuation of the psyllid population. Our study suggests positive interactions between two pests during the growing season. The positive relationship between *A. pallida* and *B. gobica* egg abundances highlights the increasing need for novel methods for Goji berry pest management. In practice, *A. pallida* control can be efficient by eliminating its vector *B. gobica*. Both pests can be controlled together, which reduces chemical usage.

## 1. Introduction

In natural and agricultural conditions, crop plants are often attacked by multiple herbivores, including gall mites and other arthropod species [1]. Interactions between the pests contribute to the suppression of plant defenses [2]. Gall mites are challenging to manage in agriculture systems due to their small size [3], high reproductive potential [4], and the ineffectiveness of available miticides [5]. As mites commonly share the same host plant and habitats as other herbivores, they can disperse by attaching to other insects, which has often been interpreted as phoresy [6,7,8]. Phoresy is a common dispersal strategy in mites. The phoront typically attaches actively to the vector, such as insects, to avoid unfavorable environments within a certain period of time [9]. Phoresy is often considered as a pattern of phoront–vector mutualism [10]. Phoronts undoubtedly benefit from phoresy (e.g., vectors provide phoronts with shelters and help dispersal), but vectors do not acquire a fitness benefit during the phoretic period, so the positive interactions may be more apparent after detachment [11]. For instance, during the growing season, some phoretic mites can remove antagonistic microorganisms or predators of their vectors from their shared habitats [12,13]. Such positive interactions promote cooperation between two pests, increasing stress in agricultural production systems.

Our previous study has found a phoretic relationship between the gall mite *Aceria pallida* Keifer (Acari: Eriophyidae) and the psyllid *Bactericera gobica* Loginova (Hemiptera: Psyllidae) [also named as *Paratrioza sinica* Yang et Li (Hemiptera: Psyllidae)] [14], which are serious pests of Goji berry *Lycium barbarum* L. (Solanaceae), one of the important economic crops in China [15]. The gall mite intentionally enters the cavity between the hindleg coxae of the psyllid as hibernation sites during the overwintering season and dismounts when the vector arrives at the host plant during the growing season [14]. Detached mites form foliar galls on Goji berry leaf surface by penetrating leaf tissues, then survive and reproduce inside the galls. Psyllid females mainly oviposit on Goji berry leaves. Their offspring almost survive and develop on the same sites after hatching because the nymphs are pretty inactive [16,17]. Thus the abundance of *B. gobica* on leaves can be represented by the plenty of its eggs [18]. Given that *A. pallida* and *B. gobica* eggs share the same habitat (i.e., Goji berry leaves), there are very likely to be interspecific interactions despite no direct contact between them. To better understand whether the interactions are facilitative or competitive, information on relationships between the abundances of *A. pallida* and *B. gobica* eggs on leaves is needed.

The phoront *A. Pallida* benefits from phoresy during the overwintering season, but no advantages to the vector *B. gobica* have been found during this period [19], which goes against the general pattern of phoront–vector mutualism. Therefore, this mutualism may occur (i.e., the vector obtains some benefits from the mite) during the growing season. To determine whether such interactions after detachment were positive or negative, we addressed three main questions in this study: 1. The relationship between gall diameter and mite abundance in the gall (which can provide a way to rapidly estimate the level of mite damage in the field by measuring the diameter of galls); 2. The mite–psyllid egg abundance relationship on Goji berry leaves; 3. Habitat selection preferences for mites and psyllids.

## 2. Materials and Methods

### 2.1. Plants and Insects

Goji berry seedlings were obtained from the Institute of Plant Protection, Ningxia Academy of Agro-Forestry Sciences, Ningxia Province, China. The seedlings were cultivated in plastic flowerpots (22 × 17 × 15 cm high) for germination in the greenhouse (25 ± 2 °C, 65 ± 5% RH, and a 16:8 h light/dark photoperiod) at the Institute of Zoology, Chinese Academy of Sciences, Beijing, China. Seedlings with approximately 30 leaves were transferred into the laboratory and selected for the study. The experiments were performed in conditions at LD 16:8, 25 ± 2 °C, 65 ± 5% RH.

Colonies of gall mite *A. pallida* and psyllid *B. gobica* were established from individuals collected in the greenhouse at the Institute of Plant Protection, Ningxia Academy of Agro-Forestry Sciences. The colonies were maintained at LD 16:8, 25 ± 2 °C, 65 ± 5% RH in the greenhouse at the Institute of Zoology, Chinese Academy of Sciences, and reared on Goji berry plants separately.

### 2.2. Relationship between Gall Diameter and A. pallida Abundance in The Gall

To better estimate the number of *A. pallida* that can easily evade visual detection, 300 galls were selected randomly from mite-infested seedlings. The diameter of galls was measured using digital calipers, and the number of mites (eggs, larvae, nymphs, and adults) living in the galls was recorded using biological microscopes.

### 2.3. Relationship between Abundances of A. pallida and B. gobica Eggs on Goji Berry Leaves

To determine the relationship between the abundances of *A. pallida* and *B. gobica* eggs on Goji berry leaves, the fresh seedlings were simultaneously infected with mite galls and psyllid eggs. Based on the methods of Westphal et al. [20], mature *A. pallida* galls were selected and cut into 2-mm-diameter slices (about 60 mites in each slice); under a stereomicroscope, three slices were carefully attached to the tip of a seedling. Subsequently, the seedling with mites was placed into a cubic mesh cage (50 × 50 × 50 cm with an 80-mesh net), including ten psyllid adults (<24 h, 1:1 sex ratio). The seedling was cultivated under the above-mentioned climate conditions. After five days, the psyllid adults were removed. All leaves of the seedling were checked, and the number of psyllid eggs and the total diameter of galls on each leaf were recorded. The experiment was replicated five times simultaneously, so a total of five seedlings were checked.

### 2.4. Habitat Selection Preferences for A. pallida and B. gobica

To determine the effect of *A. pallida* galls on oviposition habitat selection of *B. gobica*, a Goji berry seedling with galls was infested with psyllid eggs. The initial infestation with mites was conducted in a cubic mesh cage using the above-mentioned methodology. A fresh seedling without mite infestation was used as a control. Five days later, 10 psyllid adults (<24 h, 1:1 sex ratio) were introduced into the cage. The seedling was cultivated under the above-mentioned climate conditions. Three days later, we recorded the numbers of psyllid eggs on mite-infested and non-infested leaves and the lifespan of each leaf after the defoliation event. The experiments were replicated five times simultaneously; a total of ten seedlings (5 treatments and 5 controls) were tested.

A similar protocol was used with the infestation order reversed: psyllids were allowed to infest Goji berry seedlings for three days, which were then presented to mites for five days. The total diameters of galls on psyllid-infested and non-infested leaves were measured, the leaf lifespan was also recorded. A non-infested seedling was used as a control for mite habitat selection. The experiment was replicated five times simultaneously; a total ten seedlings (5 treatments and 5 controls) were examined.

### 2.5. Statistical Analysis

Descriptive statistics are given as the mean and standard error of the mean. Statistics were performed with SPSS 20.0 software (IBM, Armonk, NY, USA). Regression analyses were performed using SigmaPlot 12.0 software (Systat Software Inc., San Jose, CA, USA). Habitat selection preferences were analyzed with χ^2^ tests, with values for each combination of factors calculated based on the resulting standardized residual. Leaf lifespan data were analyzed using one-way analysis of variance (ANOVA) with Tukey’s honest significant difference (HSD) test when the data were homoscedastic or the Games−Howell test when the data were not. In all tests, *p* values < 0.05 were considered significant.

## 3. Results

### 3.1. Relationship between Diameter of Gall and A. pallida Abundance in the Gall

Diameters per gall were positively correlated with the numbers of mites per gall. The numbers with various diameters of galls fitted the linear equation well via linear regression analysis (*R*^2^ = 0.662, *F*_1,298_ = 583.136, *p* < 0.001; Figure 1). Using the linear regression gets a rule of one more millimeter on gall diameter for every 30 individuals increase in mite abundance per gall. The mean diameter of galls measured was 3.40 ± 0.11 mm, and the mean density of mites in each gall was 101.32 ± 4.24 individuals.

### 3.2. Relationship between Abundances of A. pallida and B. gobica Eggs on Goji Berry Leaves

Overall, 284 leaves on five Goji berry seedlings were measured; 31 were non-infested, 57 were only infested with mites, 46 were only infested with psyllid eggs, and 150 were infested with both mites and psyllid eggs. For each leaf infested with two pests, the total diameter of galls positively correlated with the number of psyllid eggs. Both data fit the linear equation well (*R*^2^ = 0.413, *F*_1,148_ = 105.9, *p* < 0.001; Figure 2A). The total diameter of galls per leaf increased by an average of 0.39 mm as the number of psyllid eggs in the leaf increased by one. Across the investigation period, the average total diameter of galls per leaf was 23.50 ± 1.11 mm, and the mean number of psyllid eggs per leaf was 40.42 ± 1.82 eggs.

To better describe the relationship between mite and psyllid egg abundances in a leaf, the gall diameter data were transformed into mite abundance in the gall based on the linear equation in Figure 1. For each leaf, our results showed a positive linear relationship between abundances of mites and psyllid eggs (*R*^2^ = 0.413, *F*_1,148_ = 11.235, *p* = 0.004; Figure 2B). Using the linear regression gets a rule of one more psyllid egg for every 12 mites increase. The mean gall mite abundance in a leaf was 711.61 ± 33.66 individuals.

### 3.3. Habitat Selection Preferences for A. pallida and B. gobica

We tested that 63 leaves were infested with mites for seedlings infested with mites and 94 leaves were not. For seedlings without mite infestation, 211 leaves were surveyed. Gall mite infestation had a significant effect on oviposition habitat selection of psyllid females (χ^2^_(1,N = 8834)_ = 7194.413, *p* < 0.001; Figure 3A), psyllid females preferred mite-infested leaves to non-infested leaves. For the mite-infested seedlings, the mean number of psyllid eggs on leaves without mites (mean ± SE: 3.26 ± 0.62 eggs) was more than eight times lower than that on mite-infested leaves (27.78 ± 2.51 eggs). The number of psyllid eggs on the seedlings without mite infestation was 32.12 ± 3.75 eggs.

By examining seedlings with (including 91 psyllid-infested leaves and 125 non-infested leaves) and without (including 329 non-infested leaves) psyllid infestation, we also found infestation by psyllid eggs showed a positive effect on habitat selection of gall mites (χ^2^_(1,N = 208216)_ = 192,932.891, *p* < 0.001; Figure 3B). We tested psyllid-infested seedlings and found that the number of mites living in galls on leaves without psyllid eggs (20.70 ± 4.46 individuals) was 23 times lower than that on psyllid-infested leaves (455.02 ± 20.82 individuals). The mite abundance per leaf on non-infested seedlings was 499.16 ± 38.58 individuals.

Further, we recorded the defoliation event by measuring the leaf lifespan. Overall, 140 (Mite), 39 (Mite & psyllid egg), 63 (Psyllid egg & mite), 85 (Psyllid egg), and 381 (Non-infested) leaves with different treatments were tested. The lifespan of non-infested leaves was 147.60 ± 3.22 d. By contrast, only mite infestation slightly affected the leaf lifespan (Mite = 106.86 ± 3.83 d). Mite and psyllid infestation treatments (Mite & psyllid egg = 76.36 ± 5.98 d, Psyllid egg & mite = 71.14 ± 4.18 d, regardless of whether mite was first or subsequent species to infest hosts) were characterized by a short lifespan, but still had a longer leaf lifespan than only psyllid infestation treatment (Psyllid egg = 45.26 ± 2.80 d) (*F*_4,703_ = 89.962, *p* < 0.001; Figure 3C).

## 4. Discussion

Our previous study has shown the gall mite *A. pallida* sought shelters under cold conditions as a way of overwintering strategy, so a phoretic relationship between the mite and the psyllid *B. gobica* was developed to overcome the overwintering challenge and locate proper habitats on Goji berry plants in next spring [14]. The mite can dismount when its vector arrives and reproduces on host plants as both pest species share the same hosts and habitats (i.e., Goji berry leaves). Once sharing the same habitat, interspecific interactions are inevitable between *A. pallida* and *B. gobica*. Gall insects can impact the performance of other herbivores in the same habitat by manipulating host morphology and physiology [21,22]. For instance, the gall mite *Aceria cladophthirus* Nalepa can increase the susceptibility of its host plant *Solanum dulcamara* L. to spider mite *Tetranychus urticae* Koch [23], and the gall wasp *Erynnis propertius* Scudder & Burgess adversely affects the butterfly *Neuroterus saltatorius* Edwards [24]. On the contrary, the gall midge *Rabdophaga salicisbrassicoides* Packard increases the abundance of aphids and their attendant ants [25]. Phoresy is often considered as a pattern of phoront–vector mutualism. The phoront *A. pallida* benefits from phoresy during the overwintering season, but no advantages to the vector *B. gobica* are found during this period. This study suggests this mutualism occurs during the growing season based on the outcomes of positive relationships between the abundance of these two pest species on leaves.

On Goji berry seedlings simultaneously infested by *A. pallida* and *B. gobica*, there was a positive linear relationship between the abundance of mites living in galls and psyllid eggs on a leaf. On Goji berry seedlings initially infested with psyllid eggs, gall mites preferred psyllid-infested leaves to non-infested leaves. Similarly, infestation by gall mites also had a positive impact on the oviposition habitat selection of psyllids. Our data suggest there may be indirect positive interactions between *A. pallida* and *B. gobica*, which promote each other’s growth and development.

Eggs of *B. gobica* possess an extension of the chorion called a pedicel; the egg pedicel is inserted directly into the host plant stomata [26]. In addition to “anchoring” the egg to the host plant leaf, the primary function of the egg pedicel is to serve as the primary conduit through which moisture is absorbed from the host plant [27,28]. The attachments of the fibers to the core of the pedicel suggest that the pedicel functions as the collector and conduit for water and perhaps solute movement into the egg [29], indicating *B. gobica* egg hatch may be dependent upon water uptake by the pedicel, and that the pedicel has the ability to transport solutes into the developing egg. Gall mites are prevalent in moist, humid conditions [30]. They are more likely to select leaves with a “water pump” egg pedicel to maintain a population. Besides, mites on psyllid-infested leaves can be more easily phoretic on overwintering psyllids.

Similarly, *B. gobica* was also prone to leaves infested with *A. pallida*. We found the lifespan of leaves infested with both *A. pallida* and *B. gobica* eggs was 62.9% longer than that of leaves only infested with *B. gobica* eggs. This indicates that infestation by *A. pallida* may reduce foliage defoliation caused by *B. gobica*, increasing the continuation of *B. gobica* populations. Growth phytohormones such as auxin and cytokinin secreted by gall mites can prevent leaves from senescence and abscission induced by other herbivores [31,32,33,34,35]. Thus, gall mites appear to have a favorable effect on maintaining psyllid populations. Furthermore, the damage caused by gall mites to host plants often is constructive rather than destructive partly because galls contribute to the retention of leaf water [2,36], which has been proved experimentally. Our results showed that only mite infestation had slight effects on Goji berry leaf lifespan compared to only psyllid or both species infestation. The leaf lifespan with only mite infestation was 136.1% and 44.9% longer than that with only psyllid and both species infestations, respectively. Phloem-feeding psyllids suck nutrients from foliage and cause the discoloration and ultimate abscission of host plant foliage by depleting plant resources [31,37,38]. Thus, after damage by *B. gobica*, plant foliage was more likely to senescence and resulted in a very short lifespan of only psyllid-infested leaves.

Such positive interactions not only maintain comfortable humid conditions for gall mites but also keep the number of psyllids high, which in turn helps the mites survive through winter (more easily phoretic on overwintering psyllids). The interactions promote the coexistence of the phoretic mite *A. pallida* and the vector *B. gobica* during the growing season, maintaining the continuation of both populations. Interspecific interactions are widespread among herbivores sharing similar niches, which impacts each other’s habitat selections without physical contact [39]. Gall mites, in particular, are good candidates for facilitating interactions of herbivores as they can manipulate plant metabolism and alter the quality of infested plant tissue [21,40]. Our results have demonstrated this phenomenon as *A. pallida* maintain the *B. gobica* population and increase *B. gobica* damage, so the mite control has to be a top priority. However, because gall mites are challenging to see directly, scouting generally utilizes the symptoms but not as indicators of mite abundance. As such, we provide a way for farmers to rapidly estimate mite abundance by measuring the diameter of galls on Goji berry leaves (one more millimeter on gall diameter for every 30 mites increase in a gall), contributing to quickly determining the level of damage. Our study provides a new approach for early detection and rapid response to *A. pallida* in the field.

This study indicates a positive relationship between the abundances of *A. pallida* and *B. gobica* eggs on Goji berry leaves, and both species have positive effects on each other’s habitat selections. Such interactions facilitate the coexistence of the mite and the vector during the growing season, which may further facilitate the phoretic relationship during the overwintering season. This facilitation could enhance our understanding of the ecological consequences of interspecific interactions and phoretic relationships. Currently, incomplete ecological knowledge about gall mites hinders the development of an effective integrated management program. The gall mite *A. pallida* is challenging to manage in Goji berry plants because its small size hinders visual detection, and conventional management approaches are not consistently practical. Therefore, *A. pallida* have been identified as one of the most seriously harmful pests in Goji berry production [41]. Better information about mite–psyllid interactions could provide insight that leads to developing management tactics promoting control of gall mites that can easily evade visual detection and miticides. However, the availability of such interactions needs to be confirmed with further semi-field or field experiments. The laboratory results cannot be directly extrapolated to field populations as laboratory experiments are simplified systems [42]. The positive relationship between *A. pallida* and *B. gobica* egg abundances highlights the increasing need for novel methods for mite management. In practice, *A. pallida* control and quarantine can be efficient by eliminating its vector *B. gobica*. Both pests can be controlled together, which reduces chemical usage.

## Figures and Tables

**Figure 1 insects-13-00577-f001:**
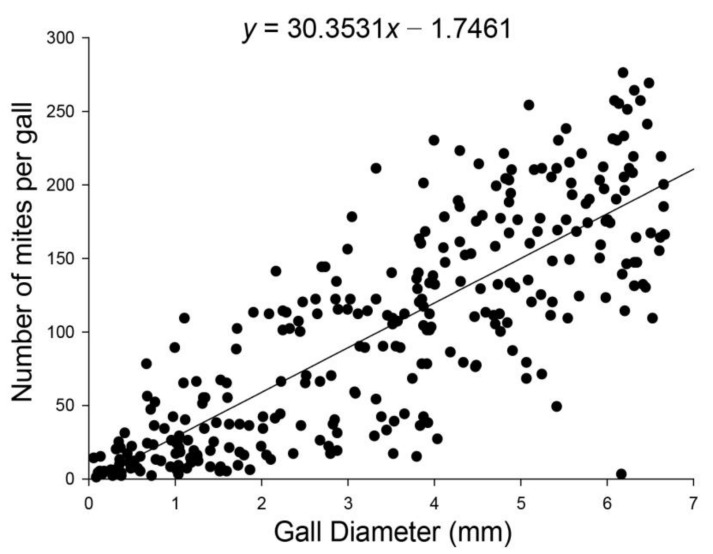
Relationship between the diameter of gall and *A. pallida* abundance in the gall. The line was fitted using linear regression analysis.

**Figure 2 insects-13-00577-f002:**
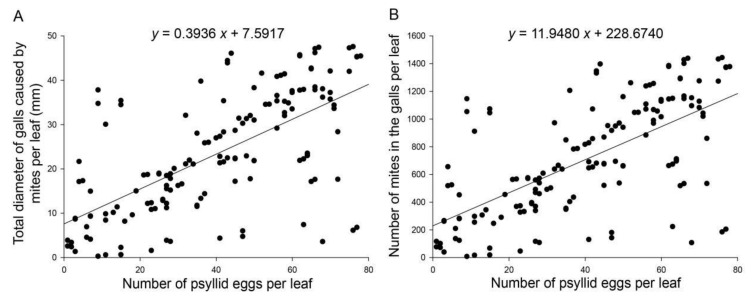
Relationship between *A. pallida* and *B. gobica* egg abundances on Goji berry leaves. (**A**) Relationship between the total diameter of galls and the number of psyllid eggs on a leaf; (**B**) Relationship between the number of mites in galls and the number of psyllid eggs on a leaf. The line was fitted using linear regression analysis.

**Figure 3 insects-13-00577-f003:**
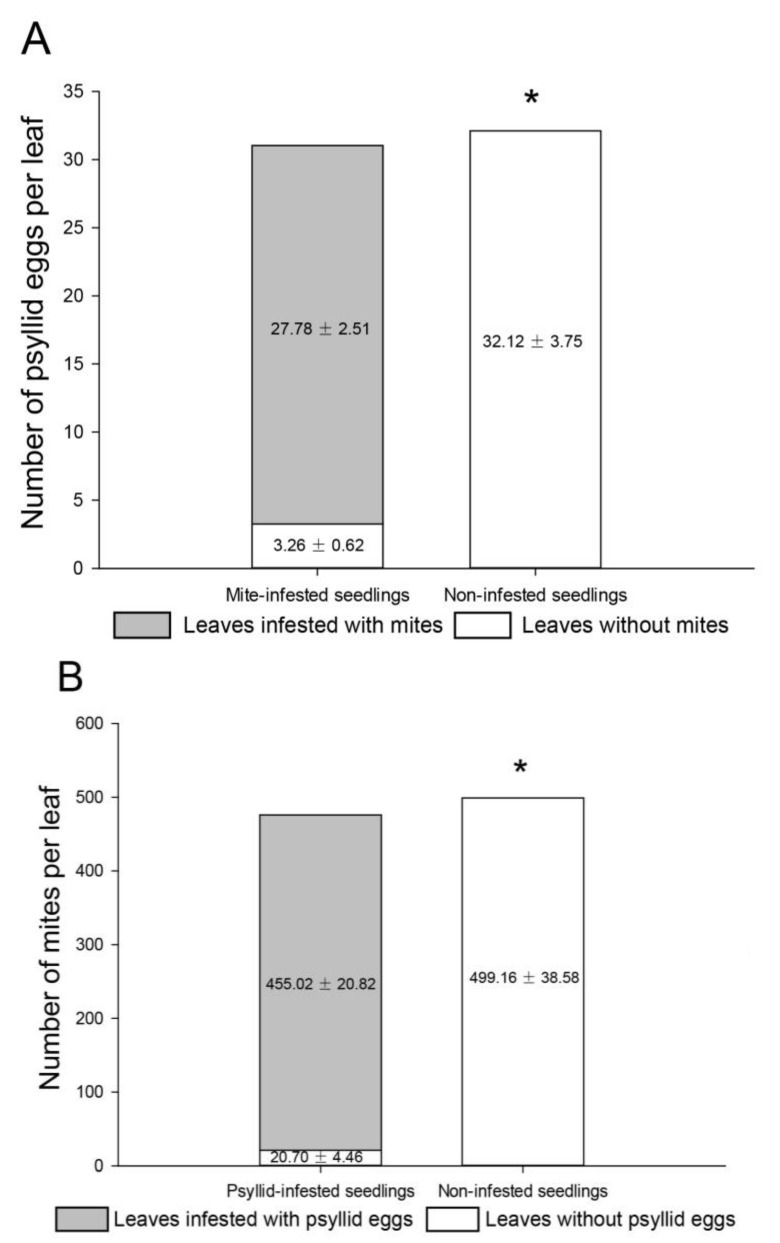
Habitat selection preferences for *A. pallida* and *B. gobica*. (**A**) On mite-infested or non-infested seedlings, mean numbers of psyllid eggs on leaves with and without mites; (**B**) On psyllid-infested or non-infested seedlings, mean numbers of mites on leaves with and without psyllid eggs. Asterisks (*) indicate a significant difference in the number of psyllid eggs (mites) on non-infested leaves between mite-infested (psyllid-infested) and non-infested seedlings (χ^2^ tests, *p* < 0.05). (**C**) The lifespan of leaves only infested with mites, leaves infested with mites and psyllid eggs (mite & psyllid egg, infested by mites first; psyllid egg & mite, invaded by psyllid eggs first), leaves only infested with psyllid eggs, and non-infested leaves. Different letters indicate significant differences among the treatments (mean ± SE; mean separation by Tukey’s HSD, *p* < 0.05).

## Data Availability

The data presented in this study are available on request from the corresponding author.

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
