# Peer review of "Positive Interactions between Aceria pallida and Bactericera gobica on Goji Berry Plants"

_insects, 2022, doi:10.3390/insects13070577_

Round 1

Reviewer 1 Report

Dear Authors, Please find attached the review of the manuscript entitled: “Relationship between the abundances of Aceria pallida and Bactericera gobica eggs on Goji berry leaves”  that has been submitted to Insects, MDPI (manuscript ID: 1763822).  The present study examines the interactions between two arthropod pest species of Goji berry - the gall mite - Aceria pallida and the psyllid - Bactericera gobica during the growing season to determine whether the coexistence is positive or negative. 

It is known that during the winter season, the mite-pest benefits from the attachment to the psyllid pest because the phoresy allows its survival in winter. However, at the same time, the phoresy does not benefit the psyllid in any way. Consequently, the authors focus on the interactions between the species during the growing season, when both pests, shortly after detachment can infest the same host plant - Goji berry. The results presented here suggest that the gall mite – psyllid coexistence during growing season is promoted by facilitative reproductive interactions, categorized as mutualism. Besides, the authors conclude that in practice, the management used against the psyllid on Goji berries should effectively reduce also the infestation by the gall mite. 

Even though the results of the study seem interesting and provide some novel insight into gall mite – psyllid coexistence, they definitely require more explanation concerning the applied methods. As far as I understood, the evidence presented came only from a laboratory experiment (or experiments? - it is very unclear) which raises serious doubts and suggests that the data collected should be treated as a preliminary. Moreover, to conclude on the psyllid pest management semi field or field experiments are required.

In my opinion, the manuscript should not be published in the present shape: the methods are not described with sufficient details, the description of results and discussion need revision. Thus, the manuscript in its current shape is unacceptable for publication and should be rejected. More details are presented below.

Title - I would suggest considering a modification to the title: “Facilitative interactions between Aceria pallida and Bactericera gobica on Goji berry plants”

Simple Summary - Should be shorter and more clear.

Abstract - Generally, the abstract reflects the content.

Keywords - I would suggest considering other words, for example: eriophyoid mites, Acariformes, Eriophyoidea, psyllid, Hemiptera, Psyllidae, coexistence, mutualism

Introduction - The first paragraph should be more comprehensive. I would suggest adding more recent studies on the interactions between eriophyoid mites and other arthropod species (for example, Glas et al. 2014. Defense suppression benefits herbivores that have a monopoly on their feeding site but can backfire within natural communities. BMC Biol. 12, 98 or Kielkiewicz et al. 2019. Unravelling the Complexity of Plant Defense Induced by a Simultaneous and Sequential Mite and Aphid Infestation. Int. J. Mol. Sci. 2019, 20, 806; doi:10.3390/ijms20040806) and concentrating on the final effect of the coexistence on the abundance of each of the pests. The purpose should be more clearly defined.

Materials and Methods -This section is written in a very unclear way. Thus, it needs improvements. I do not know how many plants were used in all the experiments. How old were they? It is not written if they were growing in controlled conditions in the climate chamber. Where were the colonies of pests maintained?

I would suggest describing ‘free-choice’ tests for pest habitat selection preference as a second paragraph following the first one concerning info on the plant and pest materials. The two types of ‘non-choice’ tests (‘A. pallida abundance in the gall and gall diameter’ and ‘B.  gobica eggs abundance’) should be described in a much clearer way as the next paragraph of this section.  

The questions concerning all experiments: How was each of the experiments designed?  How many groups of plants were used? They should be clearly named. How many plants were used as a control and mite-infested? How many plants were used as a control and psyllid-infested? How many plants were infested by mite and psyllid simultaneously and subsequently? What was the control for them? Please, avoid using the word: ‘normal’ plant.

It is not defined what the biological replication was – a plant? a leaf, or pulled leaves of one plant? Or an experiment? (line 118). What type of statistical methods were used to compare the means?

Results - Since the methods used to provide the results raise my doubts, only a corrected description of the methods would give the authors a chance to verify the results.

Discussion - The way the results were obtained does not provide enough substance for discussion. However, the discussion is the best written section. Although, numbers should not be used in this section. Conclusions are not supported by the quantitative data and statistical analysis. The information concerning further possible studies on the subject should be added.  

Author Response

Dear Authors, Please find attached the review of the manuscript entitled: “Relationship between the abundances of Aceria pallida and Bactericera gobica eggs on Goji berry leaves”that has been submitted to Insects, MDPI (manuscript ID: 1763822). The present study examines the interactions between two arthropod pest species of Goji berry - the gall mite - Aceria pallida and the psyllid - Bactericera gobica during the growing season to determine whether the coexistence is positive or negative.

It is known that during the winter season, the mite-pest benefits from the attachment to the psyllid pest because the phoresy allows its survival in winter. However, at the same time, the phoresy does not benefit the psyllid in any way. Consequently, the authors focus on the interactions between the species during the growing season, when both pests, shortly after detachment can infest the same host plant - Goji berry. The results presented here suggest that the gall mite – psyllid coexistence during growing season is promoted by facilitative reproductive interactions, categorized as mutualism. Besides, the authors conclude that in practice, the management used against the psyllid on Goji berries should effectively reduce also the infestation by the gall mite.

We are very grateful to the reviewer’s comments on our manuscript. We have studied the valuable comments carefully and tried our best to revise the manuscript. We are very sorry that we forgot to add the “2.4. Habitat selection preferences for A. pallida and B. gobica” and “2.5. Statistical analysis” in the Materials and Methods after formatting the final version, which confuses the reviewer. We have added these two important parts into manuscript to improve the readability of the paper.

Even though the results of the study seem interesting and provide some novel insight into gall mite – psyllid coexistence, they definitely require more explanation concerning the applied methods. As far as I understood, the evidence presented came only from a laboratory experiment (or experiments? - it is very unclear) which raises serious doubts and suggests that the data collected should be treated as a preliminary. Moreover, to conclude on the psyllid pest management semi field or field experiments are required.

We have added a note to the discussion section to avoid overstating our results: “However, the availability of such interactions need to be confirmed with further semi field or field experiments. The laboratory results cannot be directly extrapolated to field populations as laboratory experiments are simplified systems”. And we will certainly use field trials to further illustrate this in the future.

In my opinion, the manuscript should not be published in the present shape: the methods are not described with sufficient details, the description of results and discussion need revision. Thus, the manuscript in its current shape is unacceptable for publication and should be rejected. More details are presented below.

We are very grateful to the reviewer’s comments on our manuscript. We have studied the valuable comments carefully and tried our best to revise the manuscript.

Title - I would suggest considering a modification to the title: “Facilitative interactions between Aceria pallida and Bactericera gobica on Goji berry plants”

The title has been revised.

Simple Summary - Should be shorter and more clear.

This part has been shorter and clearer.

Abstract - Generally, the abstract reflects the content.

We have itemized our results to make it clearer.

“1) We found a positive linear relationship between the gall diameter and the mite abundance in the gall (one more millimeter on gall diameter for every 30 mites increase), which provided a way to rapidly estimate mite abundances in the field by measuring gall diameters. 2) There was a positive relationship between the abundance of mites and psyllid eggs on leaves. 3) Both species had positive effects on each other’s habitat selections.” 

Keywords - I would suggest considering other words, for example: eriophyoid mites, Acariformes, Eriophyoidea, psyllid, Hemiptera, Psyllidae, coexistence, mutualism

We modified the keywords: “Lycium barbarum L.; Eriophyid mite; Psyllid; coexistence, mutualism”

Introduction - The first paragraph should be more comprehensive. I would suggest adding more recent studies on the interactions between eriophyoid mites and other arthropod species (for example, Glas et al. 2014. Defense suppression benefits herbivores that have a monopoly on their feeding site but can backfire within natural communities. BMC Biol. 12, 98 or Kielkiewicz et al. 2019. Unravelling the Complexity of Plant Defense Induced by a Simultaneous and Sequential Mite and Aphid Infestation. Int. J. Mol. Sci. 2019, 20, 806; doi:10.3390/ijms20040806) and concentrating on the final effect of the coexistence on the abundance of each of the pests. The purpose should be more clearly defined.

The recent studies by Glas et al. 2014 and Kielkiewicz et al. 2019 have been added into the first paragraph. 

“In natural and agricultural conditions, crop plants are often attacked by multiple herbivores including gall mites and other arthropod species [1]. Interactions between the pests contribute to the suppression of plant defenses [2]. ”

Materials and Methods -This section is written in a very unclear way. Thus, it needs improvements. I do not know how many plants were used in all the experiments. How old were they? It is not written if they were growing in controlled conditions in the climate chamber. Where were the colonies of pests maintained?

We are really sorry that we forgot to add the “2.4. Habitat selection preferences for A. pallida and B. gobica” and “2.5. Statistical analysis” in the process of formatting:

2.4. Habitat selection preferences for A. pallida and B. gobica

To determine the effect of A. pallida galls on oviposition habitat selection of B. gobica, a Goji berry seedling with galls was infested with psyllid eggs. The initial infestation with mites was conducted in a cubic mesh cage using the above mentioned methodology. A fresh seedling without mite infestation was used as a control. Five days later, 10 psyllid adults (< 24 h, 1:1 sex ratio) were introduced into the cage. The seedling was cultivated under the above mentioned climate conditions. Three days later, we recorded the numbers of psyllid eggs on mite-infested and non-infested leaves, and the lifespan of each leaf after defoliation event. The experiments were replicated five times simultaneously, total ten seedlings (5 treatments and 5 controls) were tested.

A similar protocol was used with the infestation order reversed: psyllids were allowed to infest Goji berry seedlings for three days, which were then presented to mites for five days. The total diameters of galls on psyllid-infested and non-infested leaves were measured, the leaf lifespan was also recorded. A non-infested seedling was used as a control for mite habitat selection. The experiment was replicated five times simultaneously, total ten seedlings (5 treatments and 5 controls) were examined.

2.5. Statistical analysis

Descriptive statistics are given as the mean and standard error of the mean. Statistics were performed with SPSS 20.0 software (IBM, Armonk, NY). Regression analyses were performed using SigmaPlot 12.0 software (Systat Software Inc., San Jose). Habitat selection preferences were analyzed with χ2 tests, with values for each combination of factors calculated based on the resulting standardized residual. Leaf lifespan data were analyzed using one-way analysis of variance (ANOVA) with Tukey’s honest significant difference (HSD) test when the data were homoscedastic or the Games−Howell test when the data were not. In all tests, P values < 0.05 were considered significant.”

The experiments were replicated five times simultaneously, total ten seedlings (5 treatments and 5 controls) were tested. All psyllid adults used in experiments were < 24 h.

Modified: “The colonies were maintained at LD 16:8, 25 ± 2 ℃, 65% ± 5% RH in the greenhouse at the Institute of Zoology, Chinese Academy of Sciences and reared on Goji berry plants separately.” 

I would suggest describing ‘free-choice’ tests for pest habitat selection preference as a second paragraph following the first one concerning info on the plant and pest materials. The two types of ‘non-choice’ tests (‘A. pallida abundance in the gall and gall diameter’ and ‘B.  gobica eggs abundance’) should be described in a much clearer way as the next paragraph of this section.  

We added this part “2.4. Habitat selection preferences for A. pallida and B. gobica”.

The questions concerning all experiments: How was each of the experiments designed?  How many groups of plants were used? They should be clearly named. How many plants were used as a control and mite-infested? How many plants were used as a control and psyllid-infested? How many plants were infested by mite and psyllid simultaneously and subsequently? What was the control for them? Please, avoid using the word:‘normal’plant.

The experiments were replicated five times simultaneously, total ten seedlings (5 treatments and 5 controls) were tested. A fresh seedling without infestation was used as a control. We replaced “normal” with “non-infested”.  

“To determine the effect of A. pallida galls on oviposition habitat selection of B. gobica, a Goji berry seedling with galls was infested with psyllid eggs. The initial infestation with mites was conducted in a cubic mesh cage using the above mentioned methodology. A fresh seedling without mite infestation was used as a control. Five days later, 10 psyllid adults (< 24 h, 1:1 sex ratio) were introduced into the cage. The seedling was cultivated under the above mentioned climate conditions. Three days later, we recorded the numbers of psyllid eggs on mite-infested and non-infested leaves, and the lifespan of each leaf after defoliation event. The experiments were replicated five times simultaneously, total ten seedlings (5 treatments and 5 controls) were tested.

A similar protocol was used with the infestation order reversed: psyllids were allowed to infest Goji berry seedlings for three days, which were then presented to mites for five days. The total diameters of galls on psyllid-infested and non-infested leaves were measured, the leaf lifespan was also recorded. A non-infested seedling was used as a control for mite habitat selection. The experiment was replicated five times simultaneously, total ten seedlings (5 treatments and 5 controls) were examined.”

It is not defined what the biological replication was – a plant? a leaf, or pulled leaves of one plant? Or an experiment? (line 118). What type of statistical methods were used to compare the means?

A seedling is a biological replication. And we added “2.5. Statistical analysis”:

Descriptive statistics are given as the mean and standard error of the mean. Statistics were performed with SPSS 20.0 software (IBM, Armonk, NY). Regression analyses were performed using SigmaPlot 12.0 software (Systat Software Inc., San Jose). Habitat selection preferences were analyzed with χ2 tests, with values for each combination of factors calculated based on the resulting standardized residual. Leaf lifespan data were analyzed using one-way analysis of variance (ANOVA) with Tukey’s honest significant difference (HSD) test when the data were homoscedastic or the Games−Howell test when the data were not. In all tests, P values < 0.05 were considered significant.”

Results - Since the methods used to provide the results raise my doubts, only a corrected description of the methods would give the authors a chance to verify the results.

We are very sorry that we forgot to add these two important parts after formatting the final version.

Discussion - The way the results were obtained does not provide enough substance for discussion. However, the discussion is the best written section. Although, numbers should not be used in this section. Conclusions are not supported by the quantitative data and statistical analysis. The information concerning further possible studies on the subject should be added.  

We added “2.5. Statistical analysis”, which will make the data of the manuscript easier to understand.

Reviewer 2 Report

I believe this paper contributes to our knowledge of the complicated system.  There are suggested wording changes I would suggest to improve the readability of the paper.  

Line 124 Linear regression determined that for each millimeter increase in gall diameter, there was a increase in mite abundance per gall of 30 mites.

FIg1 1 x axis Gall Diameter (mm)

Lines 136 Both Data fit the linear

Line 136 and 144 - Your R squares are very, although significant for linear regressions.   This indicates that your variables are correlated but do not explain much variability in the dependent variable thus prediction value is lower.    I would add a statement similar to this and explain what that indicates for the prediction - ie wide variation in the predicted number. 

line 151 first sentence is very confusing.  We found that 63 leaves...   What do you mean by surveyed? examined?  then that is methods.  

Several places - normal is an odd way to talk about leaves without insects.  I would use the term non-infested 

Line 166 this sounds like methods again.  Start with a strong statement of results.

Author Response

I believe this paper contributes to our knowledge of the complicated system. There are suggested wording changes I would suggest to improve the readability of the paper.  

We are very grateful to the reviewer’s comments on our manuscript. We have studied the valuable comments carefully and tried our best to revise the manuscript. We are very sorry that we forgot to add the “2.4. Habitat selection preferences for A. pallida and B. gobica” and “2.5. Statistical analysis” in the Materials and Methods after formatting the final version, which confuses the reviewer. We have added these two important parts into manuscript.

Line 124 Linear regression determined that for each millimeter increase in gall diameter, there was a increase in mite abundance per gall of 30 mites.

Modified.

FIg1 1 x axis Gall Diameter (mm)

Modified.

Lines 136 Both Data fit the linear

Modified.

Line 136 and 144 - Your R squares are very, although significant for linear regressions. This indicates that your variables are correlated but do not explain much variability in the dependent variable thus prediction value is lower. I would add a statement similar to this and explain what that indicates for the prediction - ie wide variation in the predicted number.

In fact, we think the p-value is a clear indication of whether the linear relationship is significant or not. The larger the R square value is, the more significant it is. We understand what the reviewer means, which is to show the significance p value of the coefficient for better expression. The P values of these coefficients are all < 0.05.

Coefficients:

Standard Errors:

95% Confidence Intervals:

Lower Limit

Upper Limit

Fig.2A

b[0]

7.5917309

1.764043

4.1057658

11.077696

b[1]

0.393627

0.0382505

0.3180393

0.4692146

Fig.2B

b[0]

228.6740039

53.5491832

122.8542584

334.4937493

b[1]

11.9480289

1.1611295

9.6534947

14.2425631

line 151 first sentence is very confusing. We found that 63 leaves...   What do you mean by surveyed? examined?  then that is methods.  

We are really sorry that we forgot to add the materials and methods of this part in the process of  formatting. The experiment was replicated five times simultaneously, total ten seedlings (5 treatments and 5 controls) were examined. Thus for 5 treatments, we tested that 63 leaves were infested with mites for seedlings infested with mites and 94 leaves were not. For seedlings without mite infestation (5 controls), 211 leaves were surveyed.

We have added “2.4. Habitat selection preferences for A. pallida and B. gobica” and “2.5. Statistical analysis” to the materials and methods.

Several places - normal is an odd way to talk about leaves without insects.  I would use the term non-infested

We have replaced “normal” in the Fig.3 and text across the MS with “non-infested”.

Line 166 this sounds like methods again.  Start with a strong statement of results.

We think this is a general overview of our test results, which is not appropriate in the materials and methods.